# Modeling the influence of streamwise flow field acceleration on the aerodynamic performance of an actuator disc

Clemens Paul Zengler[1], Niels Troldborg[1], and Mac Gaunaa[1]

[1]Department of Wind and Energy Systems, Technical University of Denmark. Frederiksborgvej 399, 4000 Roskilde, Denmark

**Correspondence:** Clemens Paul Zengler (clezen@dtu.dk)

**Abstract.** Wind turbines operating in complex terrain can be subject to a background flow field which varies in the streamwise direction. This variation can influence the induced velocity at the turbine, and thus also the power performance. In the present work, a simple model is derived for the situation of an actuator disc (AD) operating in a background flow field featuring a constant streamwise velocity gradient. Reynolds-averaged Navier-Stokes (RANS) simulations of this scenario are performed, showing that a positive acceleration yields a reduction of induction and vice versa, a negative acceleration leads to an increase of induction. The new model accurately captures this behavior and significantly reduces the prediction error compared to classical momentum theory, where the effect of the background flow acceleration is disregarded. The model indicates that the maximum power coefficient and the corresponding values of the optimal induction and thrust coefficient depend on the flow acceleration. This possibly impacts optimal operational strategies under such conditions as well as wind turbine and wind farm design considerations, which often rely on the assumption of a streamwise uniform flow field.

## 1 Introduction

Classical momentum theory as developed by Rankine (1865), Froude (1878), and Froude (1889) yields insights into the relation between thrust and induction of an idealized wind turbine operating in a uniform flow field. Also, the theoretical limit of aerodynamic performance, referred to as the Betz-Joukowsky limit, can be derived from it (Betz, 1920; Joukowsky, 1920; Okulov and Van Kuik, 2012). The aerodynamic performance referred to here is the aerodynamic power, which is the product of thrust and axial velocity through the turbine plane. Today, momentum theory plays a fundamental role in design, power performance predictions and aeroelastic analysis of wind turbines. However, due to its simplifying assumptions, which do not necessarily apply to the actual operating conditions of modern wind turbines, various modifications to it were proposed in the past. When turbines operate in the atmospheric boundary layer, they are usually subject to wind shear, i.e. a variation of streamwise velocity over height. Various models exist to account for this. Wagner et al. (2011) suggested using a reference velocity for performance predictions based on the kinetic energy flux through the turbine disc plane and validated this model with measurements of a full-scale turbine. Later, Chamorro and Arndt (2013) derived an analytical correction of the Betz limit by including the effect of shear upstream and downstream of the wind turbine in momentum theory. Since shear would also lead to a variation of thrust force over the disc, Draper et al. (2016) included this additional effect in their model. Based on a vortex theory framework, Gaunaa et al. (2023) showed that momentum theory should be applied locally within momentum-

based design and analysis tools to properly account for the effect of shear. The problem of yaw misalignment and its effect on induction was approached in the early days by Glauert (1926) and more recently by Heck et al. (2023) and then by Tamaro et al. (2024), who incorporate both, shear and yaw misalignment in their model. Mikkelsen and Sørensen (2002) developed a correction to account for the effect of wind tunnel blockage on the aerodynamic performance of turbines in wind tunnels. The potential of using a diffuser to improve the aerodynamic turbine performance was for example studied by Jamieson (2009). Sørensen (2016) discusses the theory of both wind tunnel blockage and diffuser modeling in detail.

What has not yet been considered in modeling is the effect of a flow which is generally inhomogeneous in the streamwise direction. This can be the case in complex terrain or in dense wind farms where obstacles or turbines interact with the atmospheric boundary layer. However, various authors have shown that these variations in the streamwise flow field development can have a significant impact on the aerodynamic performance of a wind turbine, suggesting that it could also be beneficial to approach this problem from an analytical perspective. The previous literature has mainly focused on describing the phenomenon, while only few have attempted to model it. Troldborg et al. (2022) drew attention to this phenomenon in a Large Eddy Simulation (LES) study, in which they investigated the power performance of a full-scale wind turbine placed on top of the first ridge of the two ridges, which were part of the Perdigão field campaign. In the study, they varied the surface roughness and showed that this affects both the separation and the flow development behind the ridge: For low surface roughness, the flow stayed attached, while it detached for high values. However, the undisturbed wind speed at the turbine position when it was not operating remained nearly the same in all cases. Despite this, once the turbine was operating, the predicted mean power performance varied significantly between the different cases of surface roughness. This indicated, that the way the wake is advected downstream behind the turbine also affects the performance. Zengler et al. (2024) performed idealized Reynolds-averaged Navier-Stokes (RANS) simulations of an actuator disc (AD) located on top of a quasi-two dimensional Gaussian hill. Simulations were conducted for both different surface roughness and hill widths. The thrust force was distributed evenly over the AD and no turbine controller was involved, allowing a clearer picture of how exactly the performance of the AD is influenced by the flow field. Similar to Troldborg et al. (2022), they found that for low roughness the flow remained attached to the surface behind the ridge, leading to a deceleration and eventually to a higher induction at a given thrust coefficient, which is equivalent to a lower power performance. Here, the thrust coefficient was based on the total thrust due to the AD acting on the flow and the rotor-averaged wind speed at the position of the turbine when it was not operating. This approach is also adopted in the present work. With an increasing surface roughness, the situation became more complicated, especially when the undisturbed background flow detached from the surface already at the top of the ridge as was the case when the hill width was very small; in this case a deceleration was still noticeable at hub height; however, the induction would be lower compared to the flat reference case, which is equivalent to an increase in power performance. Dar et al. (2023) performed wind tunnel experiments of a more idealized scenario to study how wakes behave under streamwise pressure gradients. The pressure gradient was imposed by placing the turbine model on a ramp in the tunnel, causing the flow to accelerate or decelerate, depending on whether the ramp was tilted upward or downward. In addition, the power coefficient was reported showing an increase for an accelerating background flow and a decrease in a decelerating background flow. Revaz and Porté-Agel (2024) performed LES of a wind turbine on the ridge of a quasi-two-dimensional hill parametrized by a trigonometric function and varied geometrical

features of the setup such as hill height, hill width, plateau length, and turbine hub height. In all cases, the background flow field decelerated behind the ridge, and they reported a trend of decreasing power performance with increasing deceleration. An outlier from this trend was identified for a steep hill and a strong separation forming behind it, which is consistent with the previously mentioned findings by Zengler et al. (2024). Further works in which the influence of streamwise flow variations on power performance is reported exist (Cai et al., 2021; Mishra et al., 2024). However, it is often the case that the effect is not investigated in isolation, making it difficult to draw conclusions. For example, the presence of a controller raises the question of how this reacts in an accelerating flow field. In addition, it is often unclear what the reference wind speed is at the turbine position, which is essential to investigate the bias in power performance due to the development of the flow field downstream. Currently, most research shows that flow deceleration behind the turbine can lead to a decrease in performance, while flow acceleration results in an increase of performance.

Efforts to include the effect of acceleration in turbine modeling have been mainly focused on the wake behavior under such circumstances (Shamsoddin and Porté-Agel, 2018; Dar and Porté-Agel, 2022; Dar et al., 2023). Only Cai et al. (2021) attempted to also model the power performance in the presence of pressure gradients, which is in a sense equivalent to modeling flow acceleration. For this purpose, they used a wake model developed for pressure gradients in combination with a linearized flow solver and a control volume analysis. Their approach is based on the linear perturbation theory for the flow over hills (Jackson and Hunt, 1975; Hunt et al., 1988) and solves a Laplace equation for the disturbance in the velocity field. Although this model can predict both wake profiles and power performance, which are in sound agreement with measurements, their approach comes with an additional computational burden compared to fully analytical models and also cannot be directly incorporated into aeroelastic design as it does not yield local information about the flow state in the turbine plane. Apart from these efforts, no other work, to the authors' knowledge, has aimed at developing an analytical model to account for the effect of streamwise flow acceleration on aerodynamic performance. The present work is an attempt to model the effect of a constant streamwise flow acceleration on the induction and thus on the aerodynamic performance of a wind turbine.

The remainder of this article is structured as follows. First, a simplified model is derived to account for the effect of streamwise flow acceleration in momentum theory. Second, the model is validated with Reynolds-averaged Navier-Stokes (RANS) simulations of an actuator disc (AD) exposed to various constant velocity gradient flows. Last, implications and modeling details are discussed.

## 2 Model derivation

In the following, a model is derived which relates the induced velocity in the AD plane with the thrust coefficient of the AD in an accelerating flow field. It should be stressed that the derivation is not without contradictions from a strict, physical perspective, which will be discussed in more detail in Sec. 5.1. However, the final model proves to yield comparably accurate predictions, and the derivation itself allows for a clearer understanding of which effects the model captures. The theory development is based on the idea of modifying momentum theory to account for the fact that the underlying background flow field changes between the inlet and outlet of the streamtube surrounding the AD. The derivation is very similar to classical momentum theory

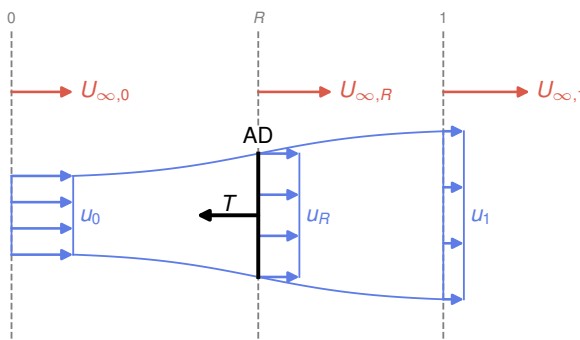

**Figure 1.** Sketch of the problem and notation for theoretical analysis.

and mainly differs when it comes to the treatment of the pressure forces acting on the streamtube and when one needs to relate the undisturbed velocities at the streamtube inlet, the AD position and the streamtube outlet with each other. The latter step is not necessary in momentum theory, since the undisturbed background flow field does not change between these positions.

As in momentum theory, a steady, incompressible, and inviscid flow past an AD is considered. Figure 1 shows a sketch of the problem and its notation, which is based on the following logic: $U$ and $u$ are the undisturbed and disturbed velocities, respectively. A subscript $_\infty$ refers to quantities in the undisturbed flow, while the subscript $_0$ denotes quantities ahead of the disc plane, $_R$ denotes quantities in the disc plane, and $_1$ denotes the ultimate wake. In addition, the subscript $_u$ denotes the case of a uniform flow field without a velocity gradient. The undisturbed flow field is subject to a constant velocity gradient, i.e.

$$\frac{\mathrm{d}U_\infty}{\mathrm{d}x} = C. \tag{1}$$

Since an infinite axial extent of the gradient would eventually lead to negative velocities up- or downstream of the AD, the extent of the stream tube considered in Fig. 1 is bounded to regions of positive velocities. The limiting case of negative velocities in the wake is discussed in Appx. B. For now, the model derivation is continued, keeping this detail in mind.

By virtue of the Bernoulli equation, the pressure jump $\Delta p$ across the AD fulfills

$$\Delta p = \frac{1}{2}\rho U_{\infty,0}^2 + p_{\infty,0} - \frac{1}{2}\rho u_1^2 - p_1. \tag{2}$$

Further, the undisturbed flow quantities ahead and behind the disc are related by the Bernoulli equation as well by

$$\frac{1}{2}\rho U_{\infty,0}^2 + p_{\infty,0} = \frac{1}{2}\rho U_{\infty,1}^2 + p_{\infty,1}, \tag{3}$$

which yields for the pressure jump

$$\Delta p = \frac{1}{2}\rho U_{\infty,1}^2 + p_{\infty,1} - \frac{1}{2}\rho u_1^2 - p_1. \tag{4}$$

By assuming, as in classical momentum theory (Rankine, 1865; Froude, 1878; Froude, 1889), that the pressure difference between the wake and the free stream is zero (i.e. $p_1 = p_{\infty,1}$) the pressure drop over the AD is seen to represent the ultimate

difference in kinetic energy between free stream and wake

$$\Delta p = \frac{1}{2}\rho \left( U_{\infty,1}^2 - u_1^2 \right).\tag{5}$$

Next, momentum balance on the stream tube sketched in Fig. 1 is considered yielding

$$\rho A_1 u_1^2 - \rho A_0 U_{\infty,0}^2 = -T + F_p,\tag{6}$$

with the thrust force $T$ related to the pressure jump as

$$T = \Delta p A_R\tag{7}$$

and the axial contribution of the pressure acting on the stream tube, $F_p$. The inlet and outlet areas of the stream tube $A_0$ and $A_1$ are related to the rotor area $A_R$ by conservation of mass

$$U_{\infty,0} A_0 = u_R A_R = u_1 A_1.\tag{8}$$

In contrast to classical momentum theory, the pressure term $F_p$ is not zero, because the pressure in the undisturbed flow is not constant in axial direction. Here, it is approximated to be

$$F_p \approx \left( p_{\infty,0} - p_1 \right) A_R.\tag{9}$$

Combining the equation for momentum balance Eq. (6) with the mass balance Eq. (8) yields

$$\rho A_R u_R u_1 - \rho A_R u_R U_{\infty,0} = -T + F_P\tag{10}$$

Further, the thrust force $T$ is replaced by the product of pressure jump and disc area as defined in Eq. (7) and the term for the pressure force (9) is introduced into the momentum balance as well

$$\rho A_R u_R u_1 - \rho A_R u_R U_{\infty,0} = -\Delta p A_R + \left( p_{\infty,0} - p_1 \right) A_R.\tag{11}$$

Expressing $\Delta p$ via Eq. (5) and $p_{\infty,0} - p_1$ via Eq. (3) yields

$$\begin{aligned}
\rho A_R u_R u_1 - \rho A_R u_R U_{\infty,0} = &-\frac{1}{2}\rho \left( U_{\infty,1}^2 - u_1^2 \right) A_R \\
&+\frac{1}{2}\rho \left( U_{\infty,1}^2 - U_{\infty,0}^2 \right) A_R.
\end{aligned}\tag{12}$$

This equation can be solved for the rotor velocity leading to

$$u_R = \frac{1}{2}\left( U_{\infty,0} + u_1 \right).\tag{13}$$

This result is similar to the classical result by Froude (1889) which says that in non-accelerating flows the velocity in the rotor plane is the average between the undisturbed, upstream velocity and wake velocity. In case of an accelerating flow, the velocity varies in the streamwise direction. Thus, it needs to be decided where to evaluate $U_\infty$ to obtain $U_{\infty,0}$, but also $U_{\infty,1}$, which

is necessary to obtain the wake velocity $u_1$ based on Eq. (5). For this purpose, a length scale $L$ is introduced, which relates the undisturbed velocity in the disc plane with the undisturbed velocities up- and downstream of the disc due to the constant gradient condition by

$$U_{\infty,0} = U_{\infty,R} - L\frac{\mathrm{d}U_{\infty}}{\mathrm{d}x}, \tag{14}$$

$$U_{\infty,1} = U_{\infty,R} + L\frac{\mathrm{d}U_{\infty}}{\mathrm{d}x}. \tag{15}$$

These relations close the presented system of equations for a given length scale. Lastly, the induction factor and the thrust coefficient are introduced as

$$a = 1 - \frac{u_R}{U_{\infty,R}}, \tag{16}$$

$$C_T = \frac{T}{\frac{1}{2}\rho A_R U_{\infty,R}^2}. \tag{17}$$

Combining Eq.(17) with Eq. (7), Eq. (5) and Eq. (13) leads to

$$\begin{aligned} C_T &= \frac{\Delta p A_R}{\frac{1}{2}\rho A_R U_{\infty,R}^2} = \frac{U_{\infty,1}^2 - u_1^2}{U_{\infty,R}^2} \\ &= \frac{U_{\infty,1}^2 - 4u_R^2 + 4u_R U_{\infty,0} - U_{\infty,0}^2}{U_{\infty,R}^2}. \end{aligned} \tag{18}$$

The free stream velocities ahead and behind the AD are replaced by the relations in Eqs. (14) and (15) and after some rearranging one obtains

$$C_T = \frac{4u_R\left(U_{\infty,R} - u_R\right) + 4\left(U_{\infty,R} - u_R\right)L\frac{\mathrm{d}U_{\infty}}{\mathrm{d}x}}{U_{\infty,R}^2}. \tag{19}$$

Applying the definition of the induction Eq. (16) and introducing the non-dimensional length scale $l = L/D$ and the non-dimensional velocity gradient $\beta = \frac{D}{U_{\infty,R}}\frac{\mathrm{d}U_{\infty}}{\mathrm{d}x}$ with D denoting the AD diameter yields the final form of the model

$$C_T = 4a(1-a) + 4al\beta. \tag{20}$$

From this equation, it can be seen that for an accelerating flow field, the relation for the thrust in a uniform flow field is modified by an additive term proportional to the free stream velocity gradient and the induction. The term $l\beta$ is equivalent to a relative change in velocity over the distance $l$, with $l$ itself being unknown. Crespo et al. (1999) stated that the length of the initial wake expansion region, thus the region where wake and free stream pressure equalize, is effectivly approximately one diameter long, a value which was also adopted by Dar and Porté-Agel (2022). Since the present derivation also assumes that the pressures equalize at the non-dimensional position $l$, a reasonable first estimate of this value correspond to one diameter, $l = 1$. The central assumptions to obtain this result are symmetry with respect to the axial extent of the control volume and the approximation of the pressure contribution to the momentum balance. Since the first component of the right-hand side of

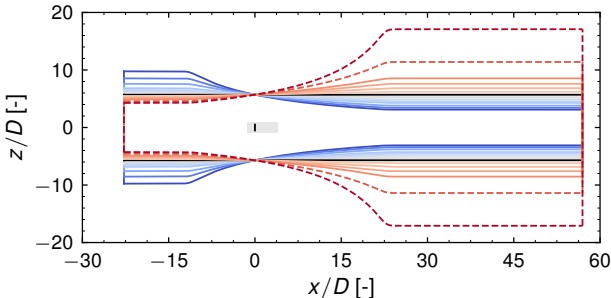

**Figure 2.** Boundaries of the computational domains considered simulating a constant velocity gradient. In the lateral direction, the domain width is 11.390 D. The gray area indicates a region of mostly cubic cells of a side length of approximately 0.047 D. Dashed lines indicates two domain configurations which are not considered within the model validation. The color-coding for the different domains is used consistently in all subsequent figures.

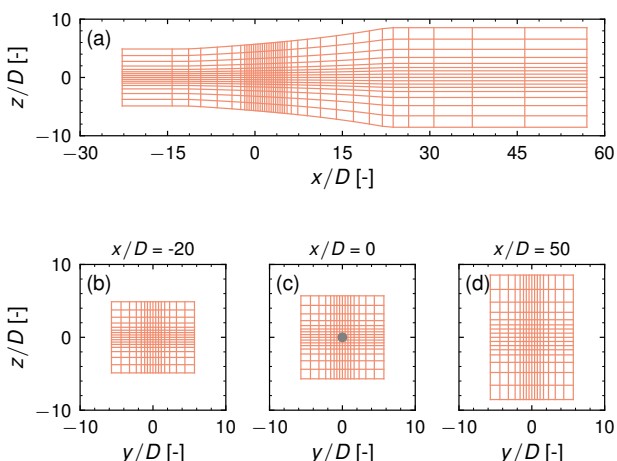

**Figure 3.** Computational grid for a deceleration, with only every eighth grid line shown. **(a)** grid in the $x$-$z$ plane, and grid in the $y$-$z$ plane at $x$ equal to **(b)** -20 D, **(c)** 0 D, and **(d)** 50 D,

Eq. (20) simply is the thrust-induction relation in a uniform flow, a reasonable generalization might be that arbitrary thrust-induction curves obtained in a uniform flow field can be corrected for acceleration by simply adding the last therm of the right-hand side of Eq. (20). Thus

$$C_T(a) = C_{T,u}(a) + 4al\beta, \tag{21}$$

with the thrust-induction relation $C_{T,u}(a)$ for a uniform background flow field. Equation (21) is the new model, which will be validated in Sec. 4.

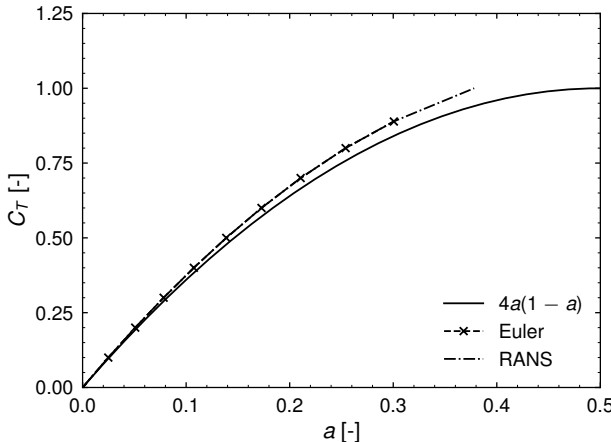

**Figure 4.** Comparison of Eulerian and RANS simulation for the baseline uniform case with theoretical results from momentum theory.

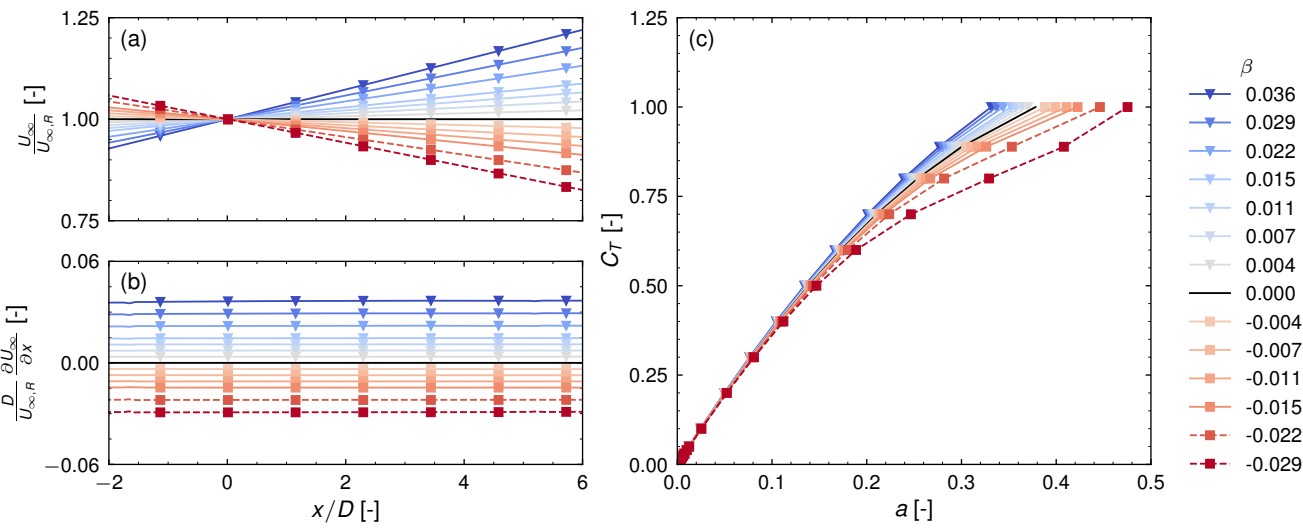

**Figure 5.** Simulation results with **(a)** the undisturbed streamwise velocity along the centerline of the AD, **(b)** the streamwise gradient, and **(c)** the predicted thrust-induction curves.

## 3   Simulation setup

Three-dimensional RANS simulations of an AD operating in a flow field featuring a constant velocity gradient in the stream-wise direction are performed in order to validate the simple model later in Sec. 4. The thrust-induction curves for a wide range of positive and negative accelerations are evaluated. In the following, the simulation setup is presented in detail.

## 3.1 Grid design

For every considered velocity gradient, a separate computational grid is generated. In all grids, the Cartesian coordinate system is centered around the AD with $x$, $y$ and $z$ denoting the streamwise, lateral and vertical directions, respectively. The cross-sectional area $A_S(x)$ of each grid is defined from mass conservation by the constant velocity gradient condition, i.e.

$$A_S(x) = \frac{U_R A_{S,R}}{U_R + x \frac{\mathrm{d} U_\infty}{\mathrm{d} x}}, \tag{22}$$

with $U_R$ and $A_{S,R}$ being the undisturbed velocity and the cross-sectional area in the AD plane. These quantities are held constant for all grids. The latter is quadratic with a side length of 11.390 D resulting in a blockage ratio of 0.605 %[1]. Based on Eq. (22), the upper and lower boundaries of the grids are modified vertically for the different velocity gradients. The lateral boundaries are not modified, resulting in a quasi-two dimensional domain. The diffuser region has a length of 34.170 D with the AD being placed at the end of its first third. Inlet and outlet regions are appended to the diffuser with a length of 11.390 D and 34.170 D, respectively, and the transitions between these regions are smoothened by a Laplacian smoothing algorithm. The vertical boundaries of the domains are shown in Fig. 2. In the situation of negative velocity gradients, it has been found that a recirculation zone can develop in the wake. In the case of the strongest negative velocity gradient simulated (later identified as the case $\beta = $ -0.029), a recirculation zone forms 8 D behind the disc at $C_T = 0.6$ and moves as close as 3 D towards the disc at $C_T = \frac{8}{9}$. In some more moderate scenarios of negative acceleration, the recirculation does not seem to affect the induction significantly, as it will be shown later in Sec. 4.1. Since recirculation is not taken into account within momentum theory and also the presented model, the cases which seem to affect the induction noticeably are disregarded in the model validation process and marked by dashed lines in Fig. 2 and all subsequent figures. A more thorough discussion on this can be found in Appx. B. Returning to the domain design, the generated grids are curvilinear and consist of $320{\times}128{\times}128$ cells in $x$, $y$, and $z$ direction, respectively. Refinement regions exist at the transitions between the diffuser and the inlet and outlet regions. Further, from a distance of 1.708 D ahead of the AD to 5.125 D behind it, a refinement region with a lateral and vertical side length of approximately 2.278 D exists. In this region, cells are nearly cubic, with a resolution of approximately 21 cells per diameter. In Fig. 3, a computational grid for the case of a deceleration (later identified as $\beta = $ -0.015) is shown.

## 3.2 Actuator disc model

The AD is simulated with a uniform distribution of thrust force per disk area. As described by Zengler et al. (2024), the outer 25 % of the radial boundaries are linearly smeared out to improve solution convergence. The induction is evaluated as the area weighted mean induction over the part of the disc, which is not affected by the smearing. The actuator shape method is applied to project the force onto the computational grid (Réthoré et al., 2014; Troldborg et al., 2015).

---

[1]The seemingly arbitrary value of 11.390 D is the consequence of the radial smearing of the AD thrust force as described by Zengler et al. (2024). If just the radial extent of the force distribution was used as reference length, the side length would be 10 D. However, to make the results comparable, the normalization diameter is calculated based on an equivalent AD with a uniform loading over the entire extent of the AD.

## 3.3 Boundary conditions, RANS closure and solver

Inlet and outlet conditions are applied at the streamwise boundaries. The inlet velocity is specified such that $U_R$ is reached in the AD plane. The chosen $U_R$ corresponds to a diameter-based Reynolds number of 3.798 million. Inlet turbulence properties correspond to a turbulence intensity of 1 % with the specific dissipation rate $\omega$ set to one million. The lateral boundaries are
periodic and slip conditions are applied at the upper and lower boundaries. As RANS closure, the two-equation $k$-$\omega$-$SST$ model without modification of the standard model coefficients is applied (Menter, 1994). The incompressible, three-dimensional, Navier-Stokes solver EllipSys3D (Sørensen, 1995; Michelsen, 1992) is employed. The used solution algorithm is a SIMPLE-like procedure based on the non-relaxed momentum equations (Sørensen, 2018). Rhie-Chow interpolation is included to avoid decoupling of the pressure and velocity fields (Rhie and Chow, 1983) and the convective terms are discretized with the QUICK
scheme (Leonard, 1979).

## 3.4 Verification and sensitivity of simulation setup

In Fig. 4, the thrust-induction curves obtained using RANS and Euler equations for a uniform background flow are compared with momentum theory. It is seen, that the Eulerian and RANS simulations do not perfectly agree with momentum theory. This discrepancy can be attributed to the numerical discretization as it has been reported in the past (Hodgson et al., 2021; Mikkelsen,
2004). If it is desired to better agree with the momentum theory results, it is necessary to increase the grid resolution or to probe the induction slightly behind the position of the AD as done in a previous study (Zengler et al., 2024). However, within this work, no correction is performed, thus the curve representing the RANS results is used as the zero-acceleration/uniform curve in the following.

In Appx. A a detailed sensitivity study of the simulation setup is performed. These results can be summarized as follows.
A mesh sensitivity analysis shows that the change of induction due to halving the cell length in all direction is less than 0.3 %. Domain blockage and the radial smearing influence the results in the order of one percent, while the effect of the diffuser length on the results is negligible.

## 4 Model validation

First, the simulation results from the previously introduced RANS simulations are presented for the various velocity gradients
considered. Subsequently, the model predictions for these cases are analyzed in detail.

## 4.1 Simulation results

Simulation results are presented in Fig. 5. The color-coding is used consistently with Fig. 2 and the subsequent Fig. 6 and Fig. 8 where each color represents one specific velocity gradient. In panels (a) and (b) of Fig. 5 the normalized streamwise velocities $U_\infty$ along the centerline and the respective normalized gradients for the different domains presented previously are
shown. In all cases, a nearly constant streamwise gradient is obtained ahead and behind the AD. Panel (c) shows the predicted

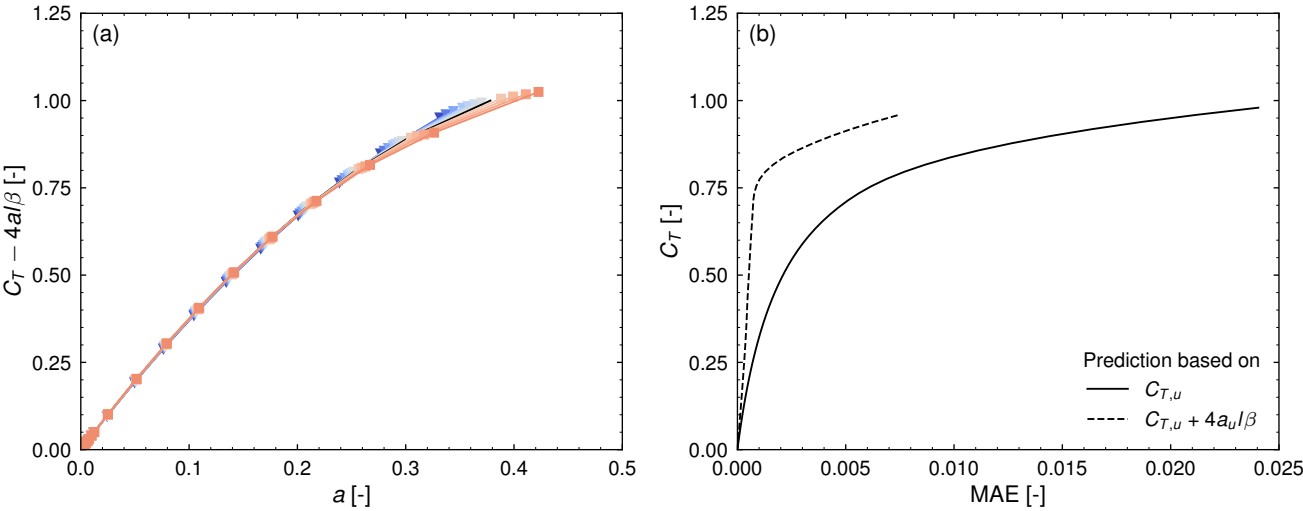

**Figure 6.** Model predictions for $l = 1$: **(a)** collapsed induction curves for $C_T - 4al\beta$ and **(b)** MAE for induction predictions without and with correction for acceleration.

thrust-induction curves obtained by applying a certain thrust coefficient and evaluating the corresponding induction. With an increasing thrust coefficient, the differences between the individual curves grow. In cases of a positive acceleration, the induction is reduced, while it is increased in cases of a negative acceleration. For the two cases with the strongest negative acceleration – dashed lines in Fig. 5 and not considered for model validation – a more complex induction behavior can be observed for thrust coefficients higher than 0.5. This can be related to the development of a recirculation zone in the wake and is discussed in Appx. B in more detail.

### 4.2 Model predictions

In the following, it is investigated to what extent the model represented by Eq. (21) is capable of predicting the trends observed in Fig. 5. The non-dimensional velocity gradient $\beta$ is evaluated in the undisturbed flow field at the position of the AD in the center, thus in Fig. 5 (b) at $x/D = 0$. As stated before, the length of the initial wake expansion region is approximately one diameter long (Crespo et al., 1999),

Model predictions are presented in Fig. 6. If the model works as expected, solving Eq. (21) for $C_{T,u}(a) = C_T(a) - 4al\beta$ and inserting the respective simulation results into the right-hand side of this formula should lead to a collapse of all simulation results on the same curve. This is presented in panel (a) of Fig. 6. Indeed, it can be seen that all curves previously shown in Fig. 5 (c) tend to collapse on the zero-acceleration curve. However, at thrust coefficients higher than 0.75, increasing discrepancies between the single curves can be observed.

To quantify the model accuracy, the mean absolute error (MAE) is evaluated as

$$\text{MAE} = \frac{\sum_{i=1}^{n} |a - a_{pred}|}{n} \tag{23}$$

over all thrust coefficients and all $n = 11$ simulations. While $a$ is the induction evaluated from a simulation at a specific accel-
eration at a given $C_T$, $a_{pred}$ is the predicted induction for this acceleration either based on the thrust-induction curve obtained
from the uniform case or from the thrust-induction curve corrected by the acceleration term. Since the power coefficient is
calculated as $C_P = C_T(1 - a)$, the MAE directly translates to a variation in $C_P$ for a given $C_T$. Results are presented in Fig. 6
(b). If no correction is considered, the mean error increases with an increasing thrust coefficient up to a value of around 0.024.
Correcting the predictions for flow acceleration yields a reduction of error for all thrust coefficients. Specifically, for $C_T < 0.5$
the error is reduced by 54 % on average, while for $C_T \geq 0.5$ the error is reduced by over 78 % on average. The maximum error
in this case at high thrust is approximately 0.007. A significant increase in the mean error is noticeable above a thrust coefficient
of 0.75. At this point, it is important to note two things. First, the MAE was only calculated for the presented simulations. It
does not yield insights into specific cases, but rather gives an estimate for the potential error reduction. Second, $l = 1$ was set
without further investigation. Based on the presented simulations, it is possible to extract the length scale, which would lead to
an exact collapse of thrust-induction curves obtained in a non-uniform flow on the uniform flow curve. This will be discussed
in Sec. 5.3.

## 5   Discussion

The above results demonstrate that the presented model captures the essential physical mechanisms influencing the induction of
an AD subject to a flow featuring a constant velocity gradient. The model structure indicates that in an accelerating flow field,
the axial induction behaves approximately as if the uniform case was subject to an additional thrust coefficient proportional to
$-4al\beta$.

### 5.1   Modeling assumptions

Within the derivation of the model, it was assumed that the pressure force acting on the streamtube can be easily quantified
as the product of the disc area and the pressure difference between upstream and downstream pressure. This assumption
eventually led to Eq. (13), which states that the rotor velocity depends on the undisturbed velocity upstream and the wake
velocity downstream. With both being dependent on the axial distance, the model results would depend on their position of
evaluation. To overcome this problem, the length scale $l$, which closes the system of equations, was introduced. For nonzero
values of $l$, model predictions depend on this free parameter. If all elements of the model were strictly correct, the arbitrary
choices of the position of the inlet and outlet of the control volume would not affect the model results, as long as they were
chosen far enough away from the disc to justify the assumption of a negligible pressure disturbance from the disc, $p_1 = p_{\infty,1}$,
used in the derivation. Further, rearranging Eq. (13) and applying Eq. (14) indicates that the wake velocity varies as follows

$$u_1 = 2u_R - U_{\infty,R} + L\frac{dU_\infty}{dx}. \tag{24}$$

For arbitrary values of $L$ respectively $l$, the energy equation yields

$$u_1 = \sqrt{U_{\infty,1}^2 - \frac{\Delta p}{\frac{1}{2}\rho}}. \tag{25}$$

According to Eq. (24), the wake velocity varies linearly with $L$. Whereas Eq. (25), which is based on first principles, shows a nonlinear dependency of $L$. These observations above reveal the effects of the inconsistencies in the assumption about the pressure force on which the model is built on.

For the used simulation setup, it was inherently assumed, that the extent of the gradient is sufficiently large, so variations of the gradient further up or downstream do not affect the induction. Or in other words, that the control volume of the model, 285   with its axial extent defined by $l$, always lies completely within the constant velocity gradient region. In the sensitivity analysis in A/ppx. A it can be seen that shortening the diffuser length 33 % has an insignificant effect on the resulting induction. For a future work, it would be interesting to investigate how short the extent of the velocity gradient can be without affecting the induction. This could shed light on understanding the length scale $l$.

The influence of the pressure force $F_p$ was in Eq. (9) assumed to be approximately the net force acting on the AD due to the 290   pressure difference up- and downstream. Its exact calculation would require the evaluation of the integral

$$F_p = - \oint_{CV} p\,d\boldsymbol{A} \cdot \boldsymbol{e_x}, \tag{26}$$

over the stream tube with the infinitesimal surface area $d\boldsymbol{A}$ and the normal vector in axial direction $\boldsymbol{e_x}$. To solve this, the exact development of the flow field within the stream tube would need to be known, which on the other hand requires the evaluation of this integral.

Despite the inconsistencies of the simple model pointed out in the discussion above, the model has shown to capture the relevant physics that an accelerating flow field has on the induction of an AD. This justifies further investigations using the model in the next section.

## 5.2   Influence on power performance

Since aerodynamic power can be generally calculated as the product of thrust and velocity, the modification of the relationship 300   between these two also yields a different power performance of wind turbines. Based on the general relation between thrust and power coefficient

$$C_P = C_T(1-a), \tag{27}$$

their curves in dependency of induction are shown in Fig. 7 for several velocity gradients. It can be readily seen, that a negative acceleration leads to a general degradation of power performance, while a positive acceleration leads to an improvement. This 305   is consistent with previous findings (Troldborg et al., 2022; Dar et al., 2023; Zengler et al., 2024; Revaz and Porté-Agel, 2024) and stresses the insight, that especially in complex terrain, the power performance gain due to terrain speed up ahead of a turbine on a hill might be reduced due to flow deceleration behind the hill.

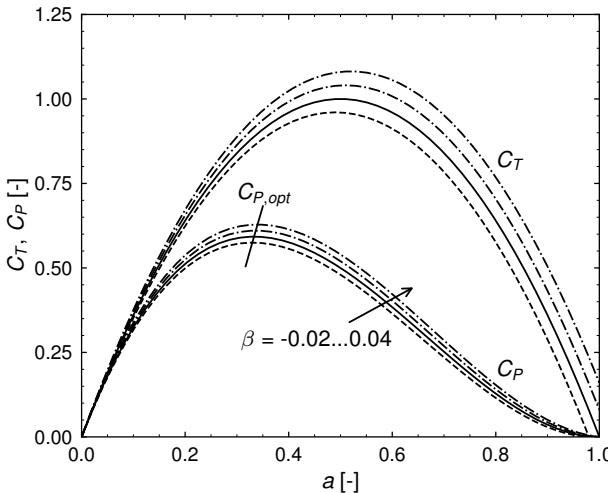

**Figure 7.** Thrust and power coefficients $C_T$ and $C_P$ based on Eq. (20) in dependency of the induction for various $\beta$ and $l = 1$. Solid lines represent the uniform background flow curve.

An obvious question is, whether the optimal operational point, thus the optimal thrust and hub height velocity are altered as well. This can be answered within the scope of the simple model by solving for the optimal induction based on Eq. (20) and Eq. (27) yielding for a constant $l$

$$a_{opt} = \frac{2}{3} + \frac{1}{3}l\beta - \frac{1}{3}\sqrt{1 + l\beta + l^2\beta^2}. \tag{28}$$

In case of a vanishing acceleration $a_{opt} = \frac{1}{3}$, which is the classical result from momentum theory. However, for a non-vanishing acceleration, the optimal induction changes with the acceleration and the length scale. This is also shown in Fig. 7. In fact, the optimum induction increases in case of a positive acceleration and decreases in case of a negative acceleration. With respect to the previously mentioned situation of a wind turbine operating on a hill, this also means that besides the physical limitation of maximum extractable energy, also optimal turbine operation is altered, which depending on the turbine controller may lead to performance degeneration relative to the highest reachable performance.

### 5.3 On the length scale

Up to now, it was assumed that the length scale $l$ has a constant value, which was adopted from literature (Crespo et al., 1999; Dar and Porté-Agel, 2022). However, it is possible to determine it for the presented simulations by solving Eq. (21) for $l$. In this context, it is important to mention that $l$ does not necessarily need to be interpreted as a physical parameter but can be seen as as a free parameter which determines how strong the influence of acceleration on the induction is. Calculating $l$ based on Eq. (21) therefore quantifies how far the model deviates from the simulated physics. This is shown in Fig. 8, where $l$ is plotted as a function of the thrust coefficient. It can be seen that $l$ depends on both the thrust coefficient and the velocity gradient, indicating that both are not sufficiently represented in the present model formulation. It is remarkable, that two separate branches of $l$

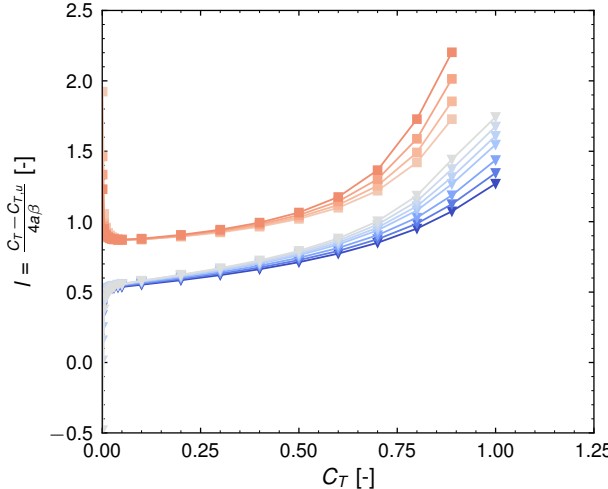

**Figure 8.** Extracted length scale $l$ from simulated thrust-induction curves.

develop, depending on whether the acceleration is negative or positive. Both branches grow towards larger values of $l$ with an increasing thrust coefficient, indicating that within the simplified modeling framework, the influence of flow field acceleration on aerodynamic performance increases with an increasing thrust coefficient. At low thrust – ignoring the two separate branches – $l$ is nearly independent of the acceleration. Very close to zero thrust, the general behavior seems to be highly non-linear, and the two branches tend to diverge from each other with the thrust coefficient approaching zero. The explanation for this behavior is not known to the authors. Note that this branching effect is not directly noticeable on the $a - C_T$ curves in Fig. 5. It should be emphasized that the length scale estimation close to zero thrust are sensitive to the grid design and numerical errors due to the small numbers involved.

The previously employed value of a reference length of one diameter (corresponding to $l = 1$) is in the range of values $l$ takes within the full range of thrust coefficients and velocity gradients. In the light of the prediction results presented earlier, this value can be seen as a compromise between low and high thrust cases.

### 5.4 Limitations and future work

As very idealized simulations were performed, a natural limitation is that the model still needs to prove its general applicability for more complex flow cases. As such, the atmospheric flow over complex terrain or the flow within a dense wind farm come to mind as mentioned earlier. However, in these scenarios, it is likely that the streamwise velocity gradient is not constant. One way to apply the presented model in these cases would be to devise a method to obtain an effective value for the combined quantity $l\beta$ from the more generally varying undisturbed background velocity.

As seen in Fig. 7, the optimal point of operation changes when a turbine is subjected to an accelerating flow field. This gives rise to the question if existing control strategies can cope with this or if it would result in the turbine operating in a suboptimal state with respect to both loads and power performance; a question which should be addressed in future works.

Lastly, only scenarios of a comparably low induction ($a < 0.5$) were investigated numerically. It is of interest how the model performs in high-induction scenarios.

## 6  Conclusions

Streamwise acceleration of the background flow field is usually not considered for wind turbine performance predictions or design of controllers. In this work, it was shown that this effect can have a significant impact on the relation between induction and thrust and therefore on aerodynamic performance predictions. A simple model, which extends classical momentum theory to the case of a constantly accelerating streamwise flow field, was presented and validated with RANS simulations. The model reduces the average prediction error in comparison with classical momentum theory for thrust coefficients lower than 0.5 by 54 % and for thrust coefficients higher than 0.5 by more than 78 %. The model indicates that the induced velocity for a given thrust depends on the background flow acceleration. The axially induced velocity decreases when the background flow has a positive streamwise acceleration, and increases for a negative streamwise acceleration. The model furthermore shows that the maximum power coefficient as well as the corresponding values of optimal induction and thrust coefficients depend on the flow acceleration. This implies that wind turbines on sites with streamwise velocity gradients may not operate optimally under such conditions. Nonetheless, the idealized conditions considered within this work do not represent realistic operating conditions of wind turbines indicating a need for further research in order to check the applicability of the presented model in practice.

*Code and data availability.* EllipSys3D used for the simulations is a proprietary software developed at DTU Wind and Energy Systems and distributed under licence. The data used in this paper are publicly available at the following DOI: https://doi.org/10.11583/DTU.27222912

## Appendix A:  Sensitivity analysis

To ensure that the presented conclusions are independent of the computational settings and geometrical choices in the RANS simulations, the sensitivity of the results with respect to cell size $dx$, blockage ratio $b$, diffuser length $L_{\text{Diff}}$, and radial smearing factor $\alpha$ at a thrust coefficient of $\frac{8}{9}$ is analyzed. All results are summarized in Tab. A1. All sensitivity investigations were carried out in a non-accelerating flow ($\beta = 0.0$), except in the diffuser length study, which assumed $\beta = -0.004$.

The influence of the cell size is studied by performing two additional simulations. One at a grid which has only half the number of cells in each dimension ($160 \times 64 \times 64$) and another one which contains twice the number of cells in each dimension ($640 \times 256 \times 256$), thus effectively halving or doubling the cells per diameter in the disc region. The first row in Tab. A1 shows that doubling the cell resolution in the disc region leads to a change of induction of 0.229 %. Based on the grid convergence

| | | | | |
|---|---|---|---|---|
| Cell size | $dx$ [D] | 0.024 | **0.047** | 0.095 |
| | $a$ [-] | 0.301 | **0.300** | 0.295 |
| | $\Delta a$ [%] | 0.229 | | -1.626 |
| Blockage ratio | $b$ [-] | 0.00151 | **0.00605** | |
| | $a$ [-] | 0.304 | **0.300** | |
| | $\Delta a$ [%] | 1.378 | | |
| Diffuser length | $L_{\mathrm{Diff}}$ [D] | 22.780 | 28.475 | **34.170** |
| | $a$ [-] | 0.305 | 0.304 | **0.305** |
| | $\Delta a$ [%] | -0.025 | -0.244 | |
| Radial smearing | $\alpha$ [-] | 0.150 | **0.250** | 0.350 |
| | $a$ [-] | 0.304 | **0.300** | 0.297 |
| | $\Delta a$ [%] | 1.143 | | -1.051 |

**Table A1.** Sensitivity of the simulation results to cell size, blockage ratio, diffuser length, and radial smearing. Bold values are used within this study.

index, the relative error with respect to the exact solution is estimated to be 0.266 % with an estimated order of convergence of $p = 2.831$ (Roache, 1994, 1997).

To investigate the blockage effect, thus the influence of the flow confinement due to the limited domain size, a disc with half
the radius of the original disc in a domain with twice the cell resolution is simulated, therefore keeping the cells per diameter constant in the disc region. The blockage ratio is defined as $b = \frac{A}{A_{S,R}}$ with the disc area $A$ and the cross-sectional area in the disc plane $A_{S,R}$. Results in Tab. A1 indicate, that reducing the blockage ratio by a factor of four leads to an increase of induction of 1.378 %.

Within the derived model, it is assumed that the velocity in the background has a constant axial gradient. In the simulations,
it is necessary to limit the axial extent of the acceleration region. To investigate if this has an influence on the simulation results, two additional simulations with a decreasing diffuser length are performed. The reference diffuser is 34.170 D long, while the two shorter ones are 28.475 D and 22.780 D long, respectively. The AD is always positioned 11.390 D behind the beginning of the diffuser. In all cases, the mildest negative velocity gradient is simulated. The change in induction, as reported in Tab. A1 is for both of the short diffusers less than one percent, indicating a negligible influence on the results. One might notice that the
variation in induction is lower for the shortest diffuser compared to the intermediate one. A possible explanation for this is that the grid in both cases is slightly different, mainly because the transition regions between the diffuser and the inlet and outlet need additional cell refinement, leading to the grid itself influencing the results here as well.

To improve solution convergence, a radial smearing factor $\alpha$ of the thrust force as described by Zengler et al. (2024) is applied in the AD model. It has a range between zero and one, where zero means that no smearing is applied while one
results in smearing along the entire radius. Besides the employed value of 0.25, simulations with 0.15 and 0.35 are conducted. Table A1 shows that the expected variation in induction is in the order of one percent. It must be noted, that changing the

smearing factor eventually leads to a variation of the blockage ratio and also of the cell resolution in the disc area, an effect which is not considered here.

Overall, it can be concluded, that the results are nearly independent of the cell size and diffuser length while domain blockage and disc smearing influence the evaluated induction in the order of one percent. Further, since the simulations results are mainly compared among each other within this study, systematic errors in the simulation setup cancel each other out in these cases.

## Appendix B: Wake breakdown

For the cases of highest negative acceleration, it can be observed that a recirculation region develops in the wake at certain thrust coefficients. This is shown in Fig. B1 where the axial velocity along the centerline at $C_T = \frac{8}{9}$ is shown. At a certain axial distance, the velocity becomes negative. This is referred to as breakdown of the wake. This observation can be roughly explained by an energy balance between the wake and the surrounding flow. Equation (5) shows that for a fixed pressure jump across the disc, the energy difference between free stream and wake velocity is constant. When the free stream velocity decrease, also the wake velocity needs to decrease. Since kinetic energy per mass scales with velocity squared, the fixed energy difference between free stream and wake results in a larger decrease of the lower wake velocity than the free stream. At a certain point, this velocity becomes zero, leading to a breakdown of the wake and the development of a recirculation zone. From Eq. (5) one can therefore derive a criterion for this situation by setting $u_1 = 0$. This result can be expressed in terms of $C_T$ as

$$\frac{U_\infty(x)}{U_{\infty,R}} < \sqrt{C_T}. \tag{B1}$$

When the normalized free stream velocity reaches a value smaller than the square root of the thrust coefficient, a breakdown of the wake can be expected. At $C_T = \frac{8}{9}$ this is the situation for $U_\infty(x) < 0.94\, U_{\infty,R}$. Comparing this value with Fig. 5, it would be expected that in the two strong deceleration cases, the velocity becomes negative around 1.94 D and 2.62 D behind the disc. Simulation results show that in both cases, this happens further downstream than predicted. This may be partly explained by the entrainment of momentum from the free stream into the wake. For future modeling, one could consider a high-thrust correction for the wake breakdown due to deceleration.

*Author contributions.* CPZ derived the model, performed the simulations and drafted the article. CPZ, NT and MG contributed to idea, theoretical analysis, methodology, and result interpretation and reviewed and edited the manuscript.

*Competing interests.* The contact author has declared that none of the authors has any competing interests.

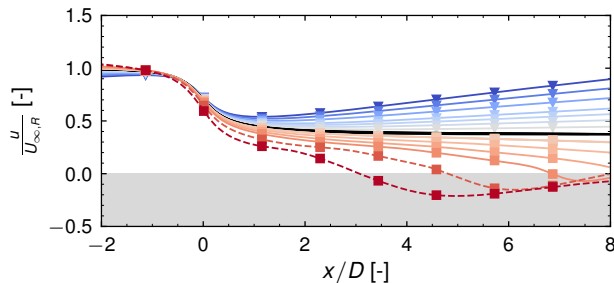

**Figure B1.** Normalized centerline velocity for $C_T = \frac{8}{9}$ for different background velocity gradients. Grey area indicates negative velocities.

*Acknowledgements.* This work has been partially supported by the EU project MERIDIONAL with grant agreement No. 101084216. We also gratefully acknowledge the computational and data resources provided on the Sophia HPC Cluster at the Technical University of Denmark (https://doi.org/10.57940/FAFC-6M81).

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
