# Peer review of "Modeling the influence of streamwise flow field acceleration on the aerodynamic performance of an actuator disc"

_Wind Energy Science, 2024_

## Referee Comment (RC2)

[referee-annotated manuscript omitted]

---

## Author Comment (AC1)

**Manuscript**
**Modelling the influence of streamwise flow field acceleration on the aerodynamic performance of an actuator disc**
**Response to reviewer 1**

Clemens Paul Zengler, Niels Troldborg, Mac Gaunaa

March 27, 2025

All changes to the manuscript can be tracked in the attached document. In the following, **bold text** is the reviewer's comments, while regular text is the author's answer.

1. **Abstract: although clearly stated in the rest of the paper, it would be more effective to also add to the abstract what is the field of application of this model (e.g., wind turbine in complex terrain);**
   The abstract was modified accordingly, at its beginning and end additional sentences were included, which contextualize the work better.

2. **Line 60: there is a typo, it should be: "U and u are the [...] velocities;**
   The respective line was corrected.

3. **The current organization of the manuscript can be improved for clarity: the description of the numerical model should be fully contained in Section 3, while a dedicated Section should be created for the validation part before the current Section 4. Figs. 3 and 4 should be moved to this new section, as they are currently too far from the part where they are commented;**
   Section 3 was renamed into 'Simulation setup' and a new section after section 3.4 called 'Model validation' introduced which features Figs. 3 and 4, which are now labelled as Figs. 5 and 6 due to new figures introduced.

4. **Figure 2: it would be nice to add a picture of the computational grid;**
   A figure (Fig. 3) was added, which represents one of the utilized grids.

5. **Figures 2b, 3, 4, 5, and 6 all have formatting issues: missing decimal separators in the axis ticks, missing legend entries, missing axis labels and units. Please revise this figures carefully;**
   This was an issue in the processing of the figures and was fixed.

6. **Line 183: a practical definition of the length scale l is stated very late in the paper. It is recommended to already discuss this aspect in Section 2, where the model is presented.**
   The manuscript was modified accordingly, and now an estimate of $l$ is already presented in Section 2.

[revised manuscript text omitted]

---

## Author Comment (AC2)

**Manuscript**
**Modelling the influence of streamwise flow field acceleration on the aerodynamic performance of an actuator disc**
**Response to reviewer 2**

Clemens Paul Zengler, Niels Troldborg, Mac Gaunaa

March 27, 2025

All changes to the manuscript can be tracked in the attached document. In the following, **bold text** is the reviewer's comments, while regular text is the author's answer.

**General remarks:**

- **The paper is quite focused. This is beneficial in some respects; however, there seems to be room for additional content. More detailed or a variation of comparison cases or a comparison with results from the literature would be valuable improvements.**
  A comparison with results from literature was originally intended to be added. However, it was difficult to identify a good comparison which would on the one side represent a comparable flow situation and on the other side provide sufficient data for an easy comparison without the need of more assumptions about the model or about the comparison data. The authors are afraid that a comparison with too many new assumptions would impair comprehensibility and clearness of the manuscript.

  For completeness, we would like to present a comparison with literature here and discuss which exact challenges arise. The work by Revaz and Porté-Agel [2024] is used for this purpose. They performed LES of a wind turbine modelled as an actuator disc including rotational effects and distributed blade forces. It was operating on a quasi-two dimensional hill described by a trigonometric function and they varied geometrical features of the setup such as hill height, hill width, length of the plateau and hub height of the turbine. Besides reporting essential turbine performance data as $C_P$, $C_T$, and $a$, they also report the non-dimensional pressure gradient of the undisturbed flow, which they estimate as $-U_\infty \frac{\mathrm{d}U_\infty}{\mathrm{d}x} \frac{D}{U_{\infty,R}^2}$. Although the available data appears suitable for validating the simple model with it, there are certain challenges involved in this, which are outlined below.

  First, in the LES, the turbine is operated with a controller and thus without knowing the details of it, it is possible that the controller itself modifies the relation between induction, thrust coefficient and power coefficient in non-uniform flow fields. This problem is difficult to avoid when comparing analytical models with high-fidelity models or measurement data. However, the information they provide is that the turbine is always operating at optimal conditions. It is not clear if by this it is referred to that the controller should theoretically always operate optimally, or if the controller is actually tuned for each respective flow case. The former appears more likely to the authors. Without questioning whether a controller is always operating optimally, we adopt this assumption in the following.

  Second, to make predictions with the presented model for situations in which not only $C_T$ and $C_P$ changes, but also the induction factor $a$ as it is the case in the work by Revaz and Porté-Agel [2024], it is necessary to know the full thrust-induction curve in a non-accelerating flow field. Because this is not provided, one needs to assume a relation, adding another layer of uncertainty to the comparison. For the present comparison and for the sake of simplicity, we assume that $C_P(a)$ is a third-order polynomial and use the knowledge about optimal induction and $C_P$ and

the assume that at an induction of zero and of one, the power performance vanishes. The function $C_T(a)$ is afterwards approximated by dividing $C_P(a)$ by the rotor velocity $(1-a)$ and a constant factor, which we assume to be the aerodynamic efficiency. The resulting curves are presented in Fig. 1 (a).

Third, for a perfect comparison, the acceleration should be constant. As this is an idealization, and for example in complex terrain it is very unlikely to find these conditions, additional research is necessary to determine how to apply the simple model in these cases. In an article which is currently being prepared, we investigate this question further. In their work, Revaz and Porté-Agel [2024] report a pressure gradient of the base flow in the near wake, which is calculated as $-U_\infty \frac{\mathrm{d}\,U_\infty}{\mathrm{d}\,x}\frac{D}{U_{\infty,R}^2}$. For the authors, it is not fully clear, how this gradient was exactly evaluated, nonetheless, a first approximation to make this data applicable to the simple model could be

$$-U_\infty \frac{\mathrm{d}\,U_\infty}{\mathrm{d}\,x}\frac{D}{U_{\infty,R}^2} \approx -\frac{\mathrm{d}\,U_{\infty,R}}{\mathrm{d}\,x}\frac{D}{U_{\infty,R}} = -\beta, \tag{1}$$

which is just a flip of the sign due to the inverse relationship of pressure gradient and velocity gradient.

One last question arising is how exactly the power coefficient should be modelled in the simple model in case of a non-ideal turbine. Following the approach of replacing $4a(1-a)$ by the thrust coefficient of a turbine and $4a(1-a)^2$ by the power coefficient, we hypothesize the following relation for the power coefficient in an accelerating flow field

$$C_P(a) = C_{P,u}(a) + C_{T,u}(a)l\beta. \tag{2}$$

Predictions are visualized in Fig. 1 (b) for two different ways of calculating $l$. In the first way, it is, as in the manuscript just set to $l=1$, in the other case the calculation of $l$ is based on the discussion in the manuscript, where it is calculated as a measure of error between the prediction and the data. Grey markers indicate cases, in which a recirculation zone developed in the wake of the turbine in the LES. Interestingly, in the RANS simulations presented in the manuscript, the development of recirculation zones was already observed for $\beta < -0.03$, which is significantly earlier than in the discussed LES and also led to the fact that the model was only validated down to $\beta \approx -0.03$. This raises the question whether the calculation of $\beta$ is consistent between both cases and which other factors might influence the wake breakdown except only the deceleration itself. Further, the spread in the LES data makes it difficult to conclude how well the model works, although the model seems to somewhat capture the measured trends, depending on what part of the data one looks at.

Because of the numerous difficulties and required approximations/guesses in the comparison, it was decided to not include any comparison with the literature and instead to develop the model further in the future to find ways to apply it consistently across different use cases. An application of the model to practical flow situations in complex terrain is currently prepared for publication in a conference proceeding.

- **Introduction/Literature: The consideration of existing literature seems superficial. This does not necessarily mean that the authors did not consider it, but as a reader, I felt the literature was listed rather perfunctorily. The discussion of this literature in the context of the proposed model is very brief.**
  The introduction was adjusted by discussing the presented literature on the effect of acceleration in more detail. Specific attention was put on literature which directly refers to the problem of the paper.

**Abstract:**

- **The abstract is short and precise, which is good from my point of view. However, adding one or two sentences on the usefulness of the new development (e.g., which realistic flow situations can now be modeled?) might increase potential readers' interest.**
  A sentence was added to the beginning of the abstract to directly draw the attention of the reader to the possible application.

[Figure]

Figure 1: **(a)** Optimum $C_P^*$ and $C_T^*$ as reported by Revaz and Porté-Agel [2024] and analytical approximations fitted to the data. **(b)** Comparison of the simple model with LES results from Revaz and Porté-Agel [2024]. Grey markers indicate cases which showed the development of a recirculation zone in the wake and are not considered in the linear fit of the data.

- **The last sentence is meaningful but indirect. It could be beneficial to state what the consequence of this finding would be for wind turbine design, for example.**
  A sentence was added to the end of the abstract. It is indeed a good point, that this would also affect turbine design based on BEM codes, since it usually relies on the classical thrust-induction relation from momentum theory. Explaining this in detail however, might be too long for an abstract.

**Specific lines:**

- **Lines 27-48: As I understand, the sentences before were related to corrections/ improvements of the momentum theory to capture the physical impacts of different flow situations. From the text, it seems a bit off-topic to cite a publication that deals with the improvement of the wind turbine performance rather than the improvement of a modelling approach.**
  The reason these publications are cited or rather that the topic of a diffuser-augmented turbine is touched upon is, that one could consider a diffuser also as a device which introduces a streamwise acceleration at and in the vicinity of the wind turbine disk. However, due to the well-defined boundary conditions in this case, it is different to model.

- **Line 43: The term 'linearized flow solver' sounds rather general to me. What kind of model is used here? This information is crucial to understand the 'computational burden' mentioned late.**
  They used a model based on linear perturbation theory developed for the flow over hills, which divides the flow into several regions. Eventually they solved a Laplace equation for the outer region, where shear stresses can be neglected. Considering the speed how fast a Laplace equation can be solved, one can certainly argue about how big this computational burden is for a single model evaluation. However, in design studies or wind farm design considerations and optimization, the additional computational burden could easily accumulate. An additional description was added to the article to ease the understanding for the reader here.

**Model derivation:**

- **To gain a better understanding, it would be helpful to describe the basic idea of the approach before delving into the mathematical derivation. I feel that there should be included that you assume a stream tube and how it is modified due to the accelerated flow field.**

The general idea is now described at the beginning of the section and it is specifically highlighted how the derivation is different from classical momentum theory.

- **From my point of view, the shape of the stream tube is modified by the accelerated flow field. As a consequence, the sketch should be modified. Or is there any reason why this should not be the case?**
  The authors agree that the shape of the stream tube is modified by the accelerated flow field. Since the exact shape depends also on the exact acceleration, for example whether it is positive or negative, it was decided to leave the shape rather generic to improve the clarity of the figure.

- **You state that the force acting on the stream tube is non-zero. On the other hand, you use the assumption of as zero pressure difference between inner and outer part of the stream tube (which is evident). Therefore, the force needs to act on the in and outlet. Is a non-zero force on the in- and outlet in accordance with your idea? Wouldn't the stream tube itself be accelerated then? Please explain.**
  The zero pressure difference assumption is only applied at the inlet and the outlet of the stream tube, and this is a zero pressure difference in the direction perpendicular to the axial/main flow direction. There exist a pressure gradient in the axial direction in everywhere in the backbone flow, which is related to the velocity gradient (application of Bernoulli shows that $\partial p_\infty/\partial x = -\rho U_\infty \partial U_\infty/\partial x$). In other words, the extent of the stream tube is defined by the streamwise position at which the pressure between stream tube and surrounding flow equalizes in the direction perpendicular to the streamwise direction. This allows to link the free stream velocity to the wake velocity. The pressure at the inlet and the outlet is different due to the acceleration of the background flow field. To further clarify this idea, the derivation of the pressure term in the model is presented below.

The exact determination of the pressure term would require the evaluation of the following integral over the surface of the stream tube

$$F_p = -\oint_{CV} p d\vec{A}\cdot\vec{e_x}. \tag{3}$$

This integral can be split into three parts, the stream tube inlet, the stream tube outlet and the sides of the stream tube, thus

$$F_p = -\oint_{A_0} p d\vec{A}\cdot\vec{e_x} - \oint_{A_S} p d\vec{A}\cdot\vec{e_x} - \oint_{A_1} p d\vec{A}\cdot\vec{e_x}. \tag{4}$$

While the integrals over $A_0$ and $A_1$ can be easily estimated, the integral over the side requires knowledge about the exact shape of the stream tube as well as the local value of the pressure at that point (which has contributions from both the disk as well as the "accelerating backbone flow"). We chose to approximate the total integral as

$$F_p \approx (p_{\infty,0} - p_1)A_R. \tag{5}$$

This relation can be obtained from the previous integral as

$$F_p \approx p_{\infty,0}A_0 - p_1 A_1 + p_{\infty,0}(A_R - A_0) - p_1(A_R - A_1). \tag{6}$$

Here the first two terms represent the contributions of the inlet and outlet, which can be considered as exact, while the last two terms can be contributed to the outer surface. In that sense, the approximation of the pressure term in the model derivation is equivalent to assuming that the inlet pressure acts on the stream tube ahead of the disc, while the outlet pressure acts on the stream tube behind the disc.

We hope this explanation clarifies the ideas behind the model.

- **Lines 100 ff.: Intermediate steps would be helpful to understand exactly what has been done here. I feel there are quite some steps to go. E.g. there is a T in equation 13 which vanished in equation 14. However, no T is given in 9,10,11 and 12. Maybe**

**T = dp\*Ar is used. It might be that I am mistaken. Please explain.**
The relationship for the thrust is mentioned in the line 80 of the manuscript, however not explicitly stated as a separate equation. Nonetheless, the authors agree with the comment, that intermediate steps are helpful, and modified the manuscript accordingly at multiple points.

- **Line 106: It seems thatL\*beta equals the relative velocity increase over a certain distance l. Naming it like this could make it easier for the reader to understand the meaning of equation 15.**
A sentence was added after Eq. 15 mentioning this.

**Specific lines:**

- **Line 165: Doesn't this contradict the sentence above? ("If it is desired to better agree with the momentum theory results, it is necessary to increase the grid resolution...")**
The formulation of grid independence might be missleading in the context of the previous sentence. By grid independence we refer in this context to two things. First, the overall conclusions of the paper are not affected by the chosen grid and second the change in induction in a local sense, thus by considering only the next level of refinement is very small ($< 0.3\%$). Nonetheless, the sentence was modified to avoid this confusion.

- **Line 207: What is the 'first-order behaviour of an actuator disc'?**
By this sentence it was meant that it seems to capture the general physics involved and correctly predicts the trends. The manuscript was modified to reflect this more detailed description.

- **It would be helpful to clearly write down which of the assumptions are inconsistent/wrong. Strictly speaking, even the assumption of an inviscid fluid is 'wrong'. However, I believe that this is not one of the assumptions which is intended to be doubted here.**
The most important assumption within the model derivation is the contribution of the pressure term since it eventually leads to an equation for the rotor velocity $u_R$ which depends on the upstream velocity and the wake velocity, with both depending on where exactly one evaluates them. The discussion was modified to highlight this fact.

**4.3 On the length scale:**

- **In the whole text, it is somehow suggested that the length scale is something like a physical parameter, that needs to be estimated. I am not quite sure if this is really the case. One could also see it as an arbitrary value that linearly influences the induction on the new model. In fact, omitting the physical derivation, one could see the whole model as a linear deviation of the induction with the flow acceleration and a second linear tuning parameter. The main assumption of the model would than be that the deviation of the induction is linearly dependent on the flow acceleration while the extent can be tuned by the length scale. Figure 6 shows that the required length scale to match the simulation results is strongly dependent on different parameters. (Right?) Therefore, the figure gives a hint on how strong the assumptions made in the model derivation differ from the (simulated) reality. The fact that there are two branches and a range of L between 0.5 and 2 for realistic operation conditions gives an hint on how strong reality differs from the proposed model. This is not a problem in my eyes. But if this is correct, it should be stated as it is.**
The authors agree with the general observation, that the length scale does not necessarily need to be a physical parameter. It is mostly meaningful in the context of a constant acceleration where it can be directly associated with a certain change in velocity over a certain distance, making it interpretable as a length scale. But for the more general case, it can be seen as a factor which influences the induction. And although the overall form of the model leaves the impression that it is purely a linearization, it is important to notice, that the exact functional form presented cannot be obtained by a simple linear expansion. This can be seen by solving the

model for the induction, which also leads to a non-linear contribution of the acceleration term to the induction. Further, the authors also agree with the observation, that Fig. 6 shows how strong the model differs from reality and which factors actually play a role. This is now also noted in the manuscript.

**Conclusion:**

- **I find the conclusion rather short. Actually, there are no conclusions on the major question of the paper: How did the proposed model work? What are the major limitations for practical use?**
  The paper is now updated, so that both questions are addressed in the conclusion.

**Further considerations based on highlighted text in the attached document**

- **Proposed formal correction of the citation of Froude and Froude in a single citation**
  Indeed, the two cited publications are by two different authors, the father William Froude [1878] and his son Robert Edmund Froude [1889], which is why the citations are listed separately.

- **Usage of the term *measured* to describe simulation results**
  The term is now changed to the verb *predicted*.

**References**

[revised manuscript text omitted]

---

## Referee Report (RR1)

Dear authors,

I appreciate that all comments have been addressed in a (mostly) satisfactory way. Of course there are still issues we might discuss in more detail. However, I see no justification to prolong the review process for this reason, as I am generally convinced that this work contains sufficient added value.

Just one last hint: I believe that the specifications of the turbine control for the validation case are defined in *Modeling turbine wakes and power losses within a wind farm using LES: An application to the Horns Rev offshore wind farm* (which is said in Revaz and Porté-Agel [2024]). For below rated wind (which is the case at optimal operation), the thrust coefficient is kept constant (see Fig. 3d), which means that there is a typical constant tip speed ratio control (the TSR is kept constant by increasing/reducing the angular speed when the wind speed is increased/reduced). This is achieved by a simple generator speed/torque relation, which is given in the same figure. In addition, I believe that the Cp-a relation is given in the dashed line in Fig. 19 of Revaz and Porté-Agel [2024].

I hope that your work can be published soon!

Best regards,
Christian W. Schulz